# Individuals with Inflammatory Bowel Disease Have an Altered Gut Microbiome Composition of Fungi and Protozoa

**DOI:** 10.3390/microorganisms10101910

**Published:** 2022-09-26

**Authors:** Gina L. Guzzo, Murthy N. Mittinty, Bastien Llamas, Jane M. Andrews, Laura S. Weyrich

**Affiliations:** 1School of Biological Sciences, University of Adelaide, Adelaide, SA 5005, Australia; 2School of Public Health, University of Adelaide, Adelaide, SA 5005, Australia; 3Indigenous Genomics Research Group, Telethon Kids Institute, Adelaide, SA 5001, Australia; 4National Centre for Indigenous Genomics, Australian National University, Canberra, ACT 2601, Australia; 5Inflammatory Bowel Disease Service, Department of Gastroenterology and Hepatology, Faculty of Health Sciences, Royal Adelaide Hospital & School of Medicine, University of Adelaide, Adelaide, SA 5000, Australia; 6Department of Anthropology and Huck Institutes of the Life Sciences, Pennsylvania State University, State College, PA 16802, USA

**Keywords:** inflammatory bowel disease (IBD), Crohn’s disease (CD), ulcerative colitis (UC), gut microbiome, fungi, protozoa, eukaryote

## Abstract

It is known that the bacterial gut microbiome is altered in inflammatory bowel disease (IBD), but far less is known about the role of eukaryotic microorganisms in IBD. While eukaryotes are rarer than bacteria within the gastrointestinal environment, the current literature suggests that they may also be implicated in IBD. In our study, we characterized these often-neglected eukaryotic microbial communities by identifying fungi and protozoa in published shotgun stool metagenomes from 355 people with IBD (206 with Crohn’s disease, 126 with ulcerative colitis, and 23 with IBD-unclassified) and 471 unaffected healthy individuals. The individuals with IBD had a higher prevalence of fungi, particularly *Saccharomyces cerevisiae*, and a lower prevalence of protozoa, particularly *Blastocystis* species (subtypes 1, 2, 3, and 4). Regression analysis showed that disease state, age, and BMI were associated with the prevalence and abundance of these two genera. We also characterized the eukaryotic gut microbiome in a shotgun stool metagenomic dataset from people with IBD who received fecal transplants, with samples pre- and post-transplantation, and from their donors. We found that in some FMT recipients, a single eukaryotic species remained stable over time, while in other recipients, the eukaryotic composition varied. We conclude that the eukaryotic gut microbiome is altered and varies over time in IBD, and future studies should aim to include these microbes when characterizing the gut microbiome in IBD.

## 1. Introduction

Inflammatory bowel disease (IBD) is a chronic relapsing-remitting condition of the gastrointestinal tract and includes Crohn’s disease (CD) and ulcerative colitis (UC) [1]. IBD results from complex interactions between the host immune system, environment, and gut microbiome [2,3,4]. The gut microbiome refers to an assemblage of gastrointestinal microorganisms, their genes and gene products, and microenvironment [5], and it has become a popular topic in IBD research due to its observed links to the etiopathology of IBD [6]. However, a consensus as to what specifically constitutes a gut microbiome signature in IBD remains lacking, due to the high variability of microbiomes between individuals and within individuals over time, and different methodologies between studies [7,8]. Nevertheless, therapies to treat IBD by targeting the gut microbiome are being developed, including faecal microbiota transplantation (FMT), even though we do not yet understand the roles that all types of microbes play in this disease [9,10,11,12]. It is likely that a better understanding of the role of the gut microbiome in IBD will enhance the development of novel diagnostics and therapies [8].

While the microbiome continues to offer an avenue for understanding and treating IBD, the majority of IBD research has focussed on intestinal bacteria [13]. Other intestinal microbes, such as eukaryotes (fungi and protozoa), are poorly studied in the context of IBD. Microbial eukaryotes make up a small proportion of gut microbiota relative to bacteria, and they are thus often overlooked or are not captured in microbiome studies [14]. For example, fungal DNA is estimated to comprise less than 1% of microbial DNA in the gut microbiome [15,16,17], and the ratio of fungi to bacteria may differ across the intestinal tract, resulting in highly variable functional contributions [16,18]. Fungi are generally thought to be pathogenic in IBD due to several reports of fungal overgrowth in the gut during disease, including increased abundances of yeasts, such as *Candida*, *Malassezia*, and *Saccharomyces* species, localisation of fungi to sites of intestinal inflammation in people with CD, and increased anti-fungal antibodies identified in IBD [17,19,20,21,22]. An overgrowth of *Candida* is hypothesised to promote intestinal inflammation and the proliferation of other opportunistic bacteria and fungi [23,24,25]. However, findings across studies of the fungal microbiome in IBD are inconsistent largely due to different methods, sample types, and an overall paucity of research [13,18,19].

Protozoa are the rarest intestinal eukaryotes and, consequently, the least studied eukaryotic types in IBD [13]. The two most common protozoa in the human gut are *Blastocystis* species and *Dientamoeba fragilis* [26,27]. While once thought to be pathogenic, more recent research suggests that these protozoa are asymptomatic colonisers of the gut, and may even be markers of health [15,28,29]. However, the commensal status of these species still remains controversial [30]. In IBD, a handful of studies have shown that affected individuals have a reduction in these protozoa compared to healthy individuals [31,32,33]. Therefore, a reduction in these species offers the prospect of being used as an indicator of IBD, and further investigation is needed to understand their prevalence in the disease and its subtypes.

In our study, we characterised the fungal and protozoan fraction of the gut microbiome in three IBD subtypes—CD, UC and IBD unclassified (IBDU)—and healthy individuals without IBD. We also identified microbial eukaryotes in a longitudinal FMT study of individuals with CD, which included samples from pre- and post-FMT time points and donor samples. We hypothesised that the eukaryotic microbiome, like the bacterial microbiome, would differ in IBD. To the best of our knowledge, this is the first large-cohort study of the eukaryotic microbiome in IBD to identify eukaryotic species using metagenomic sequencing data.

## 2. Materials and Methods

### 2.1. Study Populations and Design

We performed an observational case–control study to identify microbial eukaryotes in gut microbiome samples in publicly available datasets. This study included 826 individuals: 355 adults with IBD and 471 adult healthy control subjects (Table 1). The IBD samples were from the 1000IBD study [34] and consisted of 355 shotgun stool metagenome samples from 355 subjects with three subtypes of IBD: 206 with CD, 126 with UC and 23 with IBDU. The control samples were from the 500FG study [35] and consisted of 471 shotgun stool metagenome samples from 471 healthy subjects who did not have IBD. These two cohorts were selected for comparison for several reasons: both cohorts were from the Netherlands and both studies used the same microbial DNA extraction (Qiagen AllPrep DNA/RNA Mini Kit with mechanical lysis) and sequencing methods (whole metagenome shotgun sequencing with Illumina HiSeq).

In addition to our observational case–control study, we investigated the eukaryotic gut microbiome composition of individuals with IBD and donors in an FMT trial. The dataset included 115 longitudinal shotgun stool metagenome samples from 17 adults with CD pre- and post-FMT, and single timepoint samples from their healthy FMT donors (*n* = 5), originally published in Kong et al. [12]. All participants in this study were recruited in France. The FMT recipients achieved clinical remission via oral corticosteroids prior to receiving FMT by colonoscopy. The recipients were randomised to receive either donor FMT (*n* = 8) or a sham FMT (*n* = 9), which consisted of the transplant serum (saline solution) alone. The five donors were divided amongst the eight recipients accordingly: three donors were allocated to a single recipient each, one donor was allocated to two recipients, and the last donor was allocated to three recipients. The donor who was allocated to two recipients only had one sample present in the dataset, thus there were a total of seven donor samples instead of an expected eight. A successful outcome of FMT was defined by Sokol et al. [36] as steroid-free clinical remission at 10 weeks post-FMT.

### 2.2. Processing Metagenomic Sequences

Samples from the 1000IBD and 500FG datasets were quality-controlled with fastp version 0.20.0 [37] with parameters to trim polyG (-g) and polyX tails (-x), filter low complexity reads (-y), reduce overrepresentation of reads (-p), and correct bases in overlapping regions (-c). KneadData version 0.7.2 (https://github.com/biobakery/kneaddata, accessed on 14 August 2022), a pipeline for processing metagenomic sequencing data, was then used to remove human DNA contamination from the samples by discarding all sequencing reads that aligned to the human reference genome GRCh37/hg19 (https://www.ncbi.nlm.nih.gov/assembly/GCF_000001405.13/, accessed on 14 August 2022) with Bowtie2 [38]. The pipeline was run with settings to bypass the read trimming step (--bypass-trim) and to remove intermediate output files (--remove-intermediate-output). Before quality control, the total read count per sample in the IBD dataset ranged from 4.1 to 26.2 million sequences, with a mean read count of 10.9 million (SD: 3.8 million). After quality control, the read counts ranged from 2.7 to 25.7 million sequences per sample, reducing the mean read count to 10.6 million (SD: 3.8 million). For the control dataset, the total read count per sample before quality control ranged from 2.4 to 34.2 million sequences and had an average of 15.1 million sequences (SD: 4.1 million). After processing the sequences, the read count per sample ranged from 2.3 to 33.2 million sequences, reducing the average read count to 13.7 million (SD: 4.0 million).

The FMT study samples had previously been bioinformatically processed for sequencing quality control steps and to remove human DNA contamination, as described in [12], and we therefore did not perform these steps. The total read count per sample in the dataset ranged from 1.7 to 32.1 million sequences, with an average of 15.9 million (SD: 6.3 million) sequences per sample.

### 2.3. Identifying Eukaryotes in Gut Metagenomes

To identify eukaryotes, we used two metagenomic profiling tools: RiboTagger version 0.8.0 (https://github.com/xiechaos/ribotagger, accessed on 14 August 2022), which identifies eukaryotic DNA based on the 18S rRNA gene (v4, v5, v6, and v7 regions) [39], and EukDetect version 1.2 (https://github.com/allind/EukDetect, accessed on 14 August 2022), which identifies eukaryotic DNA by aligning sequences to a database of 521,824 microbial eukaryote marker genes [40]. Following the aforementioned sequence processing steps, RiboTagger was run on the 826 IBD and control samples with settings to identify sequences from any of the four 18S v regions (-r v4 v5 v6 v7), flags to exclude bacterial (-no-bacteria) and archaeal DNA (-no-archaea), and assigning taxonomy of putative eukaryotic DNA with the SILVA v119 database (https://www.arb-silva.de/documentation/release-119/, accessed on 14 August 2022). EukDetect was also run on the same 826 samples through the entire pipeline (--mode runall). The full EukDetect pipeline included a filtration step to remove secondary marker gene hits to reduce false positive detections [40]. After exploratory analysis of the results comparing RiboTagger and EukDetect, we decided to proceed with further analysis using only EukDetect due to its increased sensitivity for identifying eukaryotes. The 826 samples were then sorted by their FASTQ header read names (sort -k1,1 -t) to ensure matching order of forward and reverse reads and rarefied using seqtk version 1.3-r106 (https://github.com/lh3/seqtk, accessed on 14 August 2022) set to a random seed of 100 (-s100). All samples were normalised by rarefying them to 2.28 million sequences per sample, as this was the size of the smallest sample between both cohorts, ensuring that no samples were lost in the process. This rarefaction step was performed to enable abundance analysis by mitigating the effect of different sequencing library sizes on the abundances of detected eukaryotes.

We also used EukDetect to identify eukaryotes in the FMT study samples. Eukdetect was run on the samples, before and after rarefying the samples to 1.7 million sequences per sample, the size of the smallest sample (as described above). Before rarefying, 43 samples had eukaryotes detected in them, but rarefying reduced this sample size down to 13 samples. Detailed differential abundance analysis would not be possible with this small detection size, so we proceeded with unrarefied data for our analysis to maximize the recovery of any eukaryotes.

### 2.4. Statistical Analyses

#### 2.4.1. Descriptive Taxonomic Analysis

Eukaryotic distributions were examined using prevalence and abundances. Prevalence was calculated from the proportion of individuals who had a eukaryote in their sample compared to the total size of their respective cohort groupings. Abundances were equivalent to read counts of each eukaryote, post-rarefaction. The analysis of eukaryote detections was conducted in R version 4.0.2 [41] with phyloseq version 1.38.0 [42], and plots were generated with *ggplot2* version 3.3.5 [43].

#### 2.4.2. Cohort Analysis

The characteristics of the 355 IBD and 471 healthy group that were included in our regression analysis are described in Table 1. We used counts with percentages to describe categorical variables and medians with interquartile ranges to explain the distributions of continuous variables. The summary table was generated with the R package *qwraps2* version 0.5.2 [44].

Both cohorts had a higher proportion of females than males (IBD cohort female proportion: 60.28%; healthy cohort female proportion: 56.26%) and had a median BMI in the normal range. The median age of the IBD cohort was higher than the control subject group. The majority of individuals in both cohorts were non-smokers at the time of study, with 77.65% non-smokers in the IBD group and 85.88% non-smokers in the healthy group. The healthy cohort included an additional variable for smoking status—household smoker—which was not recorded for the IBD cohort. Individuals who were recorded as having a household smoker were excluded from the analysis as there was no comparator information available. The highest number of missing cases was 14 BMI values in the healthy cohort, and since this was less than 5% of the total cohort (2.97%), we did not impute values and instead performed a complete-case analysis [45].

#### 2.4.3. Regression Analysis

We performed a regression analysis to model the relationship between the abundances of eukaryotic genera found in the 355 IBD and 471 healthy groups with the demographic data of these cohorts (Table 2). Most species had too low of a prevalence to be included in the regression model; therefore, we only performed this analysis on the two most prevalent genera, *Blastocystis* and *Saccharomyces*. We used a generalised linear model (GLM), and our outcome variable was abundance, which was measured as a count variable. Usually when the outcome variable is a count variable, the family chosen for GLM is Poisson, with a *log* link function to estimate the incidence risk ratio [46]. However, many individuals in our cohort had no eukaryotes detected in their samples, and the abundance count was therefore zero for these individuals. We therefore used a zero-inflated negative binomial model (ZINB) over a Poisson to account for the surplus of zero abundance counts [47].

By using a ZINB, we assumed that an outcome of zero abundance was due to two reasons: (1) zero counts were due to technical effects, such as batch effects and inadequate sequencing depth, and (2) the eukaryote was not present in the individual (i.e., biological zero). ZINB was performed with the R package *pscl* version 1.5.5 [48] and consisted of two parts: a binary (logit) model to predict whether each zero outcome was due to technical or biological effects and a count model (negative binomial) to model the abundance counts of *Blastocystis* and *Saccharomyces*. We used a negative binomial count model instead of Poisson because we were expecting excess variation even when counts were greater than zero. To estimate robust standard errors, we used the R *sandwich* package version 3.0-1 [49]. Statistical analysis was performed in R version 4.0.2 [41].

## 3. Results

### 3.1. Detection of Eukaryotes in Gut Metagenomes

We profiled the gut microbiomes of 355 people with IBD and 471 healthy individuals with two metagenomic marker gene tools—RiboTagger and EukDetect—and found that eukaryotes were uncommon and only present in some of the individuals’ samples. Most individuals did not have any eukaryotes identified in their sample (Figure 1A). RiboTagger identified eukaryotes in only 24 of the 355 people with IBD (6.8%) and in 74 of the 471 healthy individuals (15.7%). Though EukDetect identified more eukaryotes in the individuals than RiboTagger, the proportion of individuals with a eukaryote present was still in the minority with detection of eukaryotes in 100 of the people with IBD (28.2%) and 177 healthy individuals (37.6%). The finest resolution that could be obtained with RiboTagger was at the genus level because it identifies eukaryotes solely based on the 18S rRNA region. In contrast, EukDetect’s database is better adapted to shotgun sequencing data and covers a broader range of eukaryotic diversity due to the inclusion of a large number of conserved marker genes. Therefore, EukDetect consistently identified more eukaryotic species than RiboTagger in both IBD and control subject groups, and within the IBD subtypes (Figure 1A,B; Appendix A). However, unlike RiboTagger, EukDetect did not identify *Dientamoeba* and *Galactomyces* because neither genus was present in the Eukdetect database.

Most individuals had only one eukaryotic genus detected in their sample. Ten individuals with IBD and 29 healthy individuals had more than one eukaryote detected in their sample with EukDetect, and no individuals with IBD and 14 healthy individuals had more than one eukaryote in their sample detected with RiboTaggger (Appendix A). A total of 13 eukaryotic genera were detected in the IBD group between the two marker gene tools (Figure 1C). These genera were: *Blastocystis, Candida, Clavispora*, *Cyberlindnera*, *Debaryomyces*, *Dientamoeba*, *Galactomyces*, *Malassezia*, *Meyerozyma*, *Nakaseomyces*, *Penicillium*, *Saccharomyces*, *Wickerhamomyces*. The IBDU group had the highest proportion of individuals with eukaryotes detected by EukDetect (47.8%) out of the three IBD subtypes (Figure 1B), and the proportions of individuals with eukaryotes in the CD and UC groups were similar to each other (CD: 26.2% and UC: 27.8%). The control subject group had 11 genera identified between both tools: *Blastocystis*, *Candida*, *Cyberlindnera*, *Debaryomyces*, *Dientamoeba*, *Giardia*, *Hanseniaspora*, *Malassezia*, *Penicillium*, *Pichia*, *Saccharomyces*. In total, the IBD cohort contained more fungal genera (11) than the control subject group (8). Conversely, three protozoan genera were present in the control group, whereas only one was found in the IBD group.

### 3.2. Distribution and Prevalence of Eukaryotes

#### 3.2.1. Proportional Abundances

After confirming that eukaryotes were present and detectable in the samples, we proceeded to assess shifts in the distribution of eukaryotes using the EukDetect results on rarefied samples. All samples were rarefied to 2.28 million sequences per sample in order to compare abundances and ensure that sequencing library size did not affect the abundance results. Rarefaction resulted in the loss of three genera from the detection threshold—*Giardia*, *Hanseniaspora*, and *Meyerozyma* (Appendix A)—which was a side effect expected with this technique [50,51]. The total proportional abundances for each genera showed a difference between IBD and healthy subject group compositions; most notably a high proportional abundance of *Saccharomyces* in IBD (63.1%) and high proportional abundance of *Blastocystis* in the control subjects (93.0%) (Figure 2A). The composition for CD most closely resembled the total IBD composition, with a notable similarity in *Saccharomyces* abundances (63.1% in the total IBD group and 64.9% in CD) in contrast to UC (29.6%) and IBDU (88.8%) (Figure 2B). UC had the highest abundance of *Blastocystis* (45.2%), compared to CD (16.8%) and IBDU (11.2%). Across all groups, *Blastocystis* and *Saccharomyces* were the most abundant genera.

#### 3.2.2. Species-Level Eukaryotic Prevalence

Since most individuals only had a single eukaryotic taxon in their sample, we were unable to perform detailed microbiome diversity analyses. Nevertheless, we were able to examine the distribution of eukaryotes across both cohorts at the species level due to using whole-genome marker genes with EukDetect. The IBD cohort had a greater number of species (13) than the control subject group (10) despite having a smaller overall cohort size (Figure 2C). *Candida* spp. were uncommon between both cohorts. Eight individuals with IBD (2.3%) had *Candida* spp. in their samples; two individuals (0.6%) with *C. albicans* SC5314 and six individuals (1.7%) with *C. glabrata*. Individuals with CD had most of the *Candida* spp. in the IBD cohort; seven individuals with CD (3.4%), compared to one individual with UC (0.8%) and no individuals with IBDU. In contrast, only one control subject (0.2%) had a single *Candida* species: *C. sake*, *Saccharomyces cerevisiae* S288C were notably more prevalent in the IBD group than the healthy group, being present in 31 people with IBD (8.7%) but only 13 healthy people (2.8%). *S. cerevisiae* S288C was present in 18 individuals with CD (8.7%), eight individuals with UC (6.3%), and five individuals with IBDU (21.7%). *Blastocystis* spp. were comparatively more prevalent in the healthy individuals than the individuals with IBD; 95 healthy individuals (20.2%) had at least one *Blastocystis* species in their sample whereas only 14 individuals with IBD (3.9%) had one *Blastocystis* species in their sample. *Blastocystis* spp. were present in four individuals with CD (1.9%), eight individuals with UC (6.4%) and most prevalent in two individuals with IBDU (8.7%). *Blastocystis* sp. subtype 2 was the most prevalent subtype between both cohorts—present in five individuals with IBD (1.4%) and 38 healthy individuals (8.1%).

#### 3.2.3. Shared and Unique Species

Given that eukaryotes were rare among the cohorts, we used a generous threshold and considered a species shared if it was present in at least one individual in each cohort (Figure 2D). The shared eukaryotic microbiome between all individuals with IBD and healthy individuals included all *Blastocystis* sp. subtypes (1, 2, 3, and 4), and the yeasts *Debaryomyces hansenii* CBS767 and *Saccharomyces cerevisiae* S288C. Five of the six yeast species were uniquely identified in the IBD group. Additionally, *Blastocystis hominis* was found only in the IBD group, as it was identified in a single individual with CD (Figure 2C). The three species that were unique to the control subjects, *C. sake*, *Pichia kudriavzevii*, and *Cyberlindnera jadinii* NRRL Y-1542, were rare and each only identified in three discrete individuals (Figure 2C,D).

### 3.3. Effect of Faecal Microbiota Transplantation on Eukaryotes

We detected eukaryotes in longitudinal samples from individuals with CD who received FMT and their FMT donor samples, to investigate whether FMT has an effect on the eukaryotic microbiome. Twelve of the 17 people with CD had eukaryotes detected in their sample for at least one time point; six individuals who received donor FMT and six individuals who were in the control (sham) FMT group (Appendix A). Only one of the seven donor samples had a eukaryotic species present (Figure 3). There did not appear to be a relationship between the type of species, FMT group, and FMT outcome (measured as steroid-free clinical remission at 10 weeks post-FMT) of the recipients. Several FMT recipients—recipients one, four, nine, and 15—maintained the same eukaryotic species both before and after receiving donor FMT. These individuals all experienced remission for 10 weeks post-FMT, suggesting that some eukaryotes may not interfere with success following FMT. One eukaryotic species, *Blastocystis* sp. subtype 1, was also identified in a donor sample (donor 47) but this species was not present in their recipient’s samples. Instead, their recipient (subject 15) maintained *Penicillium roqueforti* FM164 before and after FMT. Several fungal species including *P. roqueforti* FM164 were present in individuals with both success and failure outcomes—*D. hansenii* CBS767, *P. roqueforti* FM164, *P. kudriavzevii*, and *S. cerevisiae* S288C—which indicates a more complex relationship between intestinal fungi and FMT than can be resolved from this single, small study.

### 3.4. Influence of Cohort Demographics on Gut Eukaryote Composition

We performed regression analysis to model the relationship between demographics of the IBD and healthy cohorts and the prevalence and abundance of microbial eukaryotes in their gut microbiomes. The cohort characteristics summarised in Table 1 were incorporated into a zero-inflated negative binomial regression model to determine the associations of these characteristics on the abundances of *Blastocystis* and *Saccharomyces*.

Due to the low number of individuals with each species of eukaryote, we grouped the *Blastocystis* spp. identified between both cohorts (*B. hominis*, and *Blastocystis* sp. subtypes 1, 2, 3, and 4) (Figure 2C) at the genus level and modelled the association between the presence and abundance of this genera and the cohort demographics summarised in Table 1. Disease state was statistically associated with the prevalence of *Blastocystis* spp. but not the abundance of *Blastocystis* spp. (Table 2). This result indicates that only the presence of *Blastocystis* spp. was linked to a healthy state, and within those who had *Blastocystis* spp., the abundance of this genus did not statistically differ between individuals with or without IBD. BMI was associated with *Blastocystis* spp. counts; being underweight, overweight, and obese predicted a lower abundance of *Blastocystis*. Age was also negatively associated with the presence of *Blastocystis* spp., and therefore, older individuals were less likely to have *Blastocystis* spp. in their samples.

We also modelled the association between *S. cerevisiae* and the cohort demographics. In contrast to *Blastocystis* spp., we found that IBD was a predictor of *S. cerevisiae* being present and more abundant. BMI also had an association with *S. cerevisiae*, wherein being underweight predicted a lower abundance of this species. Lastly, age and sex were positively associated with *S. cerevisiae*, and an increase in age or being male predicted an increase in *S. cerevisiae* abundance.

## 4. Discussion

In this study, we explored the eukaryotic fraction of the gut microbiome in individuals with IBD and in healthy individuals without IBD. Previous studies of the eukaryotic microbiome in IBD have primarily used amplicon sequencing methods, and only one other study has examined fungi and protozoa together [13,40]. Our study is the first comprehensive investigation of intestinal eukaryotes using whole metagenome shotgun sequencing data that includes three IBD subtypes: CD, UC, and IBDU. Overall, we found that intestinal fungi and protozoa have a distinct distribution in IBD and its subtypes.

The IBD group in our study had a higher prevalence and abundance of fungi—notably *Saccharomyces*, *Nakaseomyces*, *Debaryomyces*, and *Candida* species—compared to the healthy cohort (Figure 2). Previous findings of fungal diversity in IBD have varied [19,52], but two studies have reported a higher abundance of *Candida* species in adults with IBD [25,53]. While *Candida* spp. were more prevalent in the IBD group in our study, they were only present in eight of the 355 total individuals with IBD. Further, a *Candida* species (*C. sake*) was only present in one of the healthy individuals, making it difficult to ascertain the overall relationship between *Candida* spp. and disease state. While this low prevalence of *Candida* was unexpected, the source studies of our datasets did not specifically target eukaryotes in their methodologies, which likely resulted in the loss of eukaryotic DNA from some species (discussed subsequently). Therefore, the absence of eukaryotic species in this study cannot be directly compared to previous studies.

We also found that *Saccharomyces cerevisiae* was more prevalent in all three IBD subtypes than in the healthy individuals and was statistically associated with IBD. This finding has not been reported in other microbiome studies, although one previous study reported the opposite finding: a decreased abundance of *S. cerevisiae* in active IBD [53]. However, we did not compare active and inactive IBD in our study, and differences between disease states would almost certainly affect the fungal microbiome [53,54,55]. Likewise, we did not have data on the current therapies, particularly immunosuppressants and steroids, and dietary composition of the IBD and control groups, factors that have been shown to affect intestinal fungal composition [56,57,58,59]. A larger sample size would also be required to stratify for these variables within the groups. While our study shows associative trends between fungi and IBD, we could not ascertain whether there may be any causal relationship between these microbes and the disease. This would entail a larger, longitudinal study with sampling of the microbiome in high-risk individuals prior to their development of IBD, such as the GEM project (www.gemproject.ca, accessed on 14 August 2022). It is a complex endeavour however, as the fungal microbiome is known to also be related to diet, BMI, and other lifestyle choices (e.g., smoking, alcohol, common medications such as proton pump inhibitors and antibiotics) and controlling for such a large variety of factors over a long period of time necessitates either a large sample size or a large effect size [56,57,60,61,62,63,64].

The fact that *S. cerevisiae* was more prevalent in individuals with IBD may have implications for microbial-based therapies, such as probiotics. A popular probiotic *S. cerevisiae* strain, *S. cerevisiae* var. *boulardii*, has been trialled for various gastrointestinal disorders [65] and is generally regarded as safe, though there have been several reports of adverse events [66,67,68]. This strain has been trialled for IBD as well, but its safety and efficacy for the disease remain inconclusive [69]. While we only detected the strain *S. cerevisiae* S288C in our study, that may be because it was the only strain of *S. cerevisiae* included in the EukDetect database. The two strains share over 99% genome sequence similarity [70] and, therefore, individuals with IBD may be more susceptible to colonisation or adverse effects of *S. cerevisiae* strains. More research is needed to understand the eukaryotic microbiome in IBD to ensure the safety of probiotic strains such as *S. boulardii*.

In comparing IBD to a cohort of 471 unaffected individuals, we were able to characterise the healthy eukaryotic microbiome as well. Protozoa, namely *Blastocystis* and *Dientamoeba*, were notably more common in the healthy cohort than in IBD and associated with younger individuals with a healthy BMI (Figure 1 and Figure 2, Table 2). Our findings confirm previous findings that *Blastocystis* spp. and *Dientamoeba* are more prevalent in the healthy gut microbiome than in IBD [31,32,71]. These protozoa have been historically attributed to gastroenteritis, but research in recent years, including our present study, is causing this paradigm to shift towards considering *Blastocystis* and *Dientamoeba* as common commensals of the healthy gut microbiome [28,29,33,72,73,74]. *Blastocystis* are specifically associated with higher bacterial diversity in healthy individuals [28,33]. This observation has been explained by *Blastocystis* functioning in a predator-prey relationship wherein they compete with opportunistic bacterial pathosymbionts and prevent them from overgrowing [28,75]. Therefore, the loss of this microbe may contribute to the reduced bacterial diversity commonly observed in IBD. Patients with gastroenteritis symptoms are occasionally prescribed antibiotics when *Blastocystis* or *Dientamoeba* are detected, which may further drive disturbance of the microbiome [76,77]. Given the new evidence, we encourage clinicians to now reconsider their practice in this regard and refrain from prescribing antibiotics due to the potential for long-term microbiome perturbations. Further, the recommendations for FMT in Australia do not support exclusion of donors based on the presence of *Blastocystis* and *Dientamoeba* [78]. Our findings add further support to this recommendation.

Comparing cohorts from different studies can inflate differences observed between groups [79], and we therefore tried to mitigate the biases that this approach introduces by only comparing studies that used similar methods and by keeping the studies that used different methods separate in our analyses. For example, it is known that geographical location has an influence on the composition of an individual’s microbiome due to differences in environmental and lifestyle factors [80]. We controlled for this confounding effect by comparing two cohorts from the same country (Netherlands) in part of our study, and this comparison was kept separate from our analysis of the individuals in the FMT study who were recruited in France. The eukaryotic microbiome composition may differ in people with IBD in other regions of the world, akin to what has been observed in healthy individuals [81], and further research is needed that includes many individuals from a diversity of geographical locations.

Unlike geography, we could not properly age-match the participants between our studies, and there was a notable age difference between the two cohorts (Table 1). It is known that age can affect the diversity and composition of the gut microbiome [82], including intestinal fungi which can differ in diversity and composition from infancy to adulthood [83]. However, little is known about how fungal populations change throughout adulthood as previous work has not compared older adults to younger ones [84,85]. In our study, we found that increased age in adulthood was associated with an increased abundance of *S. cerevisiae* (Table 2), and this trend may also occur in other fungi we observed. We also found that older adults were less likely to have the *Blastocystis* spp. present in their gut microbiome, indicating that protozoa should be a point of inclusion in future age-stratified research.

We also chose two studies that used the same DNA extraction and sequencing methods since these can affect the resulting composition of metagenomic sequences [79]. However, we could not control for other biases introduced from different laboratory settings, such as different sample collection and reagents [79]. Another issue we could not account for was contamination, as neither study published negative controls (e.g., environmental or DNA extraction blanks) to measure background levels of contaminant DNA. Contamination is normally considered a greater issue for low microbial biomass studies than for high biomass studies such as those involving stool samples [86]. However, the rarity of eukaryotes means that their detection may be more affected by contamination. Future research on the eukaryotic microbiome should endeavour to include negative controls where possible.

We also examined the longitudinal eukaryotic microbiome in individuals with CD who had received FMT, as well as single-timepoint samples from their FMT donors. Eukaryotes were variable and inconsistent across time between and within the FMT recipients (Figure 3). A high variability of eukaryotes over time has been observed in other studies, and fungi specifically show a higher intra- and inter-individual variability than bacteria in healthy individuals [15,87]. Only a subset of recipients maintained the same eukaryotic species over time, and this was irrespective of treatment group or outcome. Several recipients in the active FMT group maintained the same eukaryotic species before and after FMT, suggesting that the eukaryotic microbiome may remain stable even with FMT intervention in certain individuals. We expected to observe engraftment of donor eukaryotes in their recipients’ microbiomes, but we did not observe this. This could be due to the small cohort size and would indicate that eukaryotic engraftment occurs at a low rate. However, the initial FMT study was not designed specifically for eukaryotic metagenomics and this likely limited our ability to detect eukaryotes. Although we did detect a single eukaryotic species, *Blastocystis* sp. subtype 1, in one donor, this species was not present in their recipient’s samples. Interestingly, the recipient achieved a successful FMT outcome, which further supports investigating whether *Blastocystis* spp. are indeed safe to donate in FMT [78].

Across all the study groups, eukaryotes were rare or absent in the majority of samples (Figure 1 and Figure 3). This finding was not surprising given that eukaryotes comprise a small proportion (less than 1%) of the total DNA of the gut microbiome [15,16,17]. Thus, deeper sequencing or alternative methodologies may be required to effectively detect eukaryotic DNA [18]. Additionally, all three of the source studies of our datasets were originally designed for investigating the bacterial microbiome and did not use methods to specifically preserve eukaryotic DNA for sequencing. Eukaryotic DNA can be enriched during microbial DNA extraction to maximise the likelihood of capturing eukaryotes in microbiome samples [14]. This may have biased our results towards more robust eukaryotic species whose DNA was not destroyed during the bacterial DNA extraction process, as mechanical lysis has been shown to significantly reduce fungal DNA yield compared to no lysis [14]. Despite these limitations, our results demonstrate that the eukaryotic microbiome can still be gleaned from metagenomic samples in some cases even when they are not enriched for eukaryotic DNA.

Since eukaryotes have begun to attract attention for their importance in gut microbiome research, we had several bioinformatic tools at our disposal to search for microbial eukaryotes [13]. We chose to use two different pipelines, RiboTagger and EukDetect, for their ease of use, lower demand for computational resources relative to other tools available, and the diversity provided using two separate databases [39,40]. RiboTagger identifies eukaryotes based on the 18S rRNA marker gene, whereas EukDetect’s database includes 521,824 universal eukaryotic marker genes not limited to the 18S gene. Thus, we expected to identify more eukaryotes with greater resolution with EukDetect because eukaryotes could be identified by more than just their 18S genes. Indeed, EukDetect captured an overall higher prevalence and diversity of eukaryotes than RiboTagger (Figure 1). However, unlike RiboTagger, EukDetect failed to capture *Dientamoeba* and *Galactomyces*. *Dientamoeba* was unable to be detected as its genome has never been sequenced and is absent from EukDetect’s database [40]. *Galactomyces* was also absent from EukDetect’s database. The differing results between these two tools highlights how databases may limit findings based on how they were curated. Thus, we recommend using more than one database when using marker gene detection. Additionally, most available databases primarily consist of species that have been cultured and sequenced [88], and future work would benefit from identifying uncultured microbial eukaryotes via de novo genome assembly [89]. We were also unable to explore strain variation within the species we identified in this study because we only used marker gene profilers due to computational limitations. Strain-level variation could further explain differences between IBD subtypes and healthy individuals, and it would be beneficial for future research to include this analysis.

## 5. Conclusions

The aim of our study was to explore the fungal and protozoan fraction of the gut microbiome in IBD. We were able to elucidate the IBD eukaryotic microbiome with greater precision than previous studies by using whole metagenome sequencing data to identify eukaryotic at the species level. Although these approaches need significant advancements in the future, we found that IBD and its subtypes have a eukaryotic microbiome composition distinct from individuals without IBD. Our findings highlight the need for more research that explores the nonbacterial microbiome in IBD, particularly studies that aim to preserve or enrich eukaryotic DNA and account for various factors that can affect the host eukaryotic microbiome (e.g., diet, medications, disease activity) [18,62,64]. Additionally, the results of our longitudinal analysis indicate that the eukaryotic microbiome varies over time, and future longitudinal sampling will be important to reveal the dynamics of intestinal eukaryotes. Our results also support the growing body of literature suggesting that *Blastocystis* are common in healthy individuals and associate with indicators of health. By including eukaryotes in our study, we provide a more comprehensive understanding of what constitutes a healthy microbiome from a diseased state such as IBD, and our results supplement findings on the bacterial microbiome in IBD. We advocate that a more inclusive approach to microbiome research not limited to bacteria is increasingly important as diagnostics and therapeutics for IBD are continuing to target the microbiome.

## Figures and Tables

**Figure 1 microorganisms-10-01910-f001:**
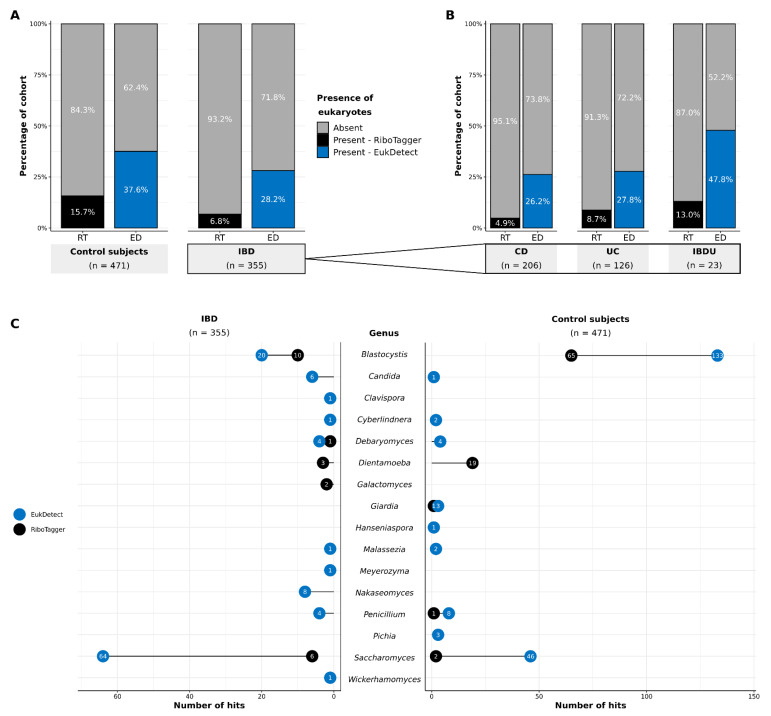
Prevalence of microbial eukaryotes in shotgun stool metagenomes of people with inflammatory bowel disease (IBD) (*n* = 355) and unaffected individuals (control subjects) (*n* = 471). (**A**) The proportion of individuals with microbial eukaryotes in their sample with at least family-level resolution is shown according to results detected by RiboTagger (RT) and EukDetect (ED) in control subject and IBD cohorts; (**B**) The same results are displayed as a proportion of samples with microbial eukaryotes detected in individuals with their IBD subtype—Crohn’s disease (CD) (*n* = 206), ulcerative colitis (UC) (*n* = 126), and IBD unclassified (IBDU) (*n* = 23)—of the total IBD cohort; (**C**) The detection of microbial eukaryotes with at least genus-level resolution by RiboTagger and EukDetect in people with IBD and control subjects. A hit represents a single taxon identified in one sample. Individuals in all groups had only one sample per individual. Results are based on unrarefied sequencing data.

**Figure 2 microorganisms-10-01910-f002:**
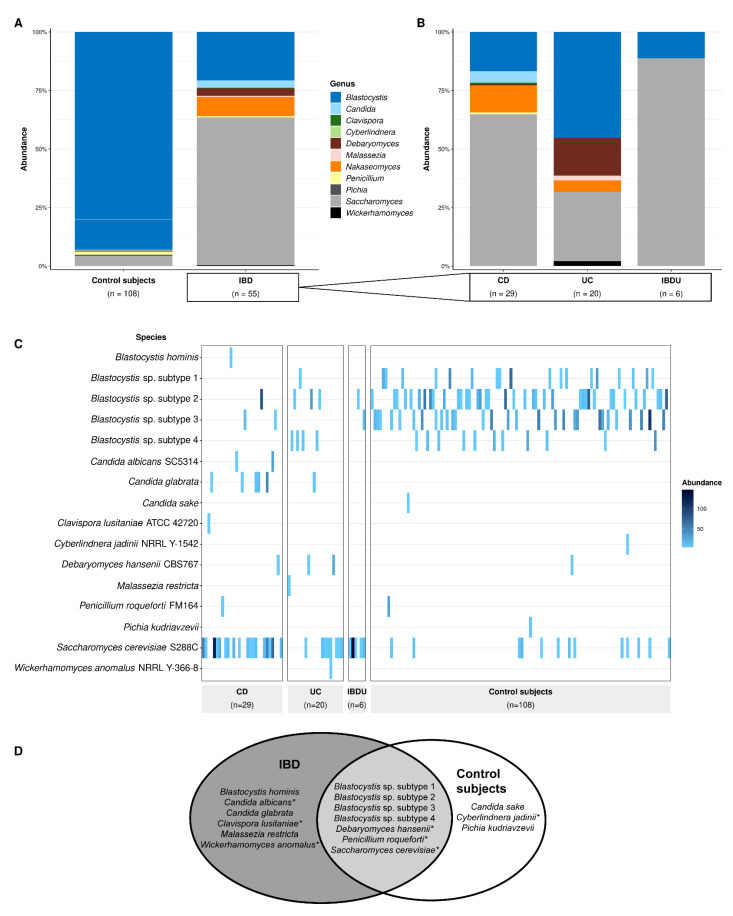
The prevalence and distribution of microbial eukaryotes in shotgun stool metagenomes is shown in people with inflammatory bowel disease (IBD) (*n* = 355) and unaffected healthy people (control subjects) (*n* = 471). (**A**) Total proportional abundances of microbial eukaryotes detected at the species-level in control subjects (*n* = 108) and individuals with IBD (*n* = 55); (**B**) Total proportional abundances are also broken down for all the people with IBD into each of their IBD subtypes—Crohn’s disease (CD) (*n* = 29), ulcerative colitis (UC) (*n* = 20), and IBD unclassified (IBDU) (*n* = 6); (**C**) Distribution and abundances of microbial eukaryotes detected at the species-level in people with IBD, specified by their IBD subtype, compared to the control subject group (*n* = 108); (**D**) Eukaryotic species unique to IBD and control subject groups and shared between groups. A species was considered shared if it was found in at least one individual in the IBD group and one individual in the control subject group. * Complete species names are: *Candida albicans* SC5314, *Clavispora lusitaniae* ATCC 42720, *Cyberlindnera jadinii* NRRL Y-1542, *Debaryomyces hansenii* CBS767, *Penicillium roqueforti FM164*, *Saccharomyces cerevisiae* S288C, and *Wickerhamomyces anomalus* NRRL Y-366-8. Results are based on EukDetect identifications in sequencing data rarefied to 2.28 million sequences per sample.

**Figure 3 microorganisms-10-01910-f003:**
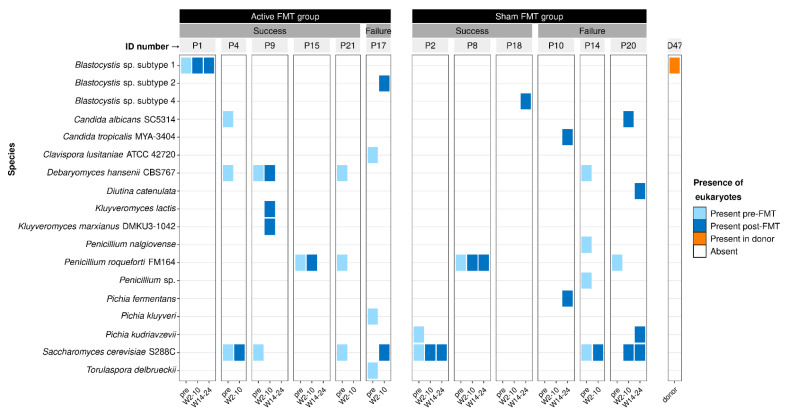
Longitudinal prevalence and distribution of microbial eukaryotes detected in shotgun stool metagenome samples from people with Crohn’s disease (CD) who received faecal microbiota transplants (FMT). Eukaryotes were also identified in a single donor sample (donor 47) whose recipient was patient 15. FMT recipients in the Active FMT group received a donor sample, and recipients in the Sham FMT group received a sample of transplant serum (saline solution) only. Success refers to steroid-free clinical remission at ten weeks post-FMT. Individuals are identified by their ID numbers, with P preceding patient ID’s and D preceding the donor ID. ‘Pre’ is pre-FMT and includes two weeks before and day of FMT, ‘W2-10’ includes results from weeks 2, 6, and 10 post-FMT, and ‘W14-24’ includes results from weeks 14, 18, and 24 post-FMT. Recipients 4, 14, 17, and 21 did not have any samples for weeks 14–24. Results are based on EukDetect identifications in unrarefied sequencing data.

**Table 1 microorganisms-10-01910-t001:** Summary characteristics of the study groups included in regression modelling.

Cohort Characteristics		IBD Group (*n* = 355)	Control Group (*n* = 471)
**Sex**	Female	214 (60.28%)	265 (56.26%)
	Male	141 (39.72%)	200 (42.46%)
	Unspecified	0 (0.00%)	6 (1.27%)
**Age (years)**	Median (IQR)	45.00 (34.25, 59.00)	23.00 (21.00, 27.00)
	18–40	145 (41.43%)	408 (87.74%)
	41–60	132 (37.71%)	31 (6.67%)
	61–80	69 (19.71%)	26 (5.59%)
	81+	4 (1.14%)	0 (0.00%)
	Missing	5	6
**BMI**	Median (IQR)	24.80 (21.70, 28.10)	22.30 (20.72, 24.39)
	Missing	0	14
**Smoking Status**	Current	78 (22.35%)	60 (14.12%)
	Past	146 (41.83%)	65 (15.29%)
	Never	125 (35.82%)	300 (70.59%)
	Missing	6	46
**Diagnosis**	CD	206 (58.03%)	NA
	UC	126 (35.49%)	
	IBDU	23 (6.48%)	

**Table 2 microorganisms-10-01910-t002:** Regression analysis results.

Model	Coefficient	*Blastocystis* spp. ^1^			*Saccharomyces cerevisiae*		
		Estimate	Std. Error	*p*-Value	Estimate	Std. Error	*p*-Value
**Count**	IBD	−0.34	0.39	0.38	1.05	0.34	**<0.01**
	Sex—male	0.23	0.22	0.29	−1.15	0.41	**0.01**
	Age	0.01	0.01	0.35	−0.03	0.01	**<0.01**
	BMI—underweight	−0.85	0.40	**0.04**	−2.19	0.61	**<0.01**
	BMI—overweight	−0.81	0.28	**<0.01**	−0.69	0.45	0.13
	BMI—obese	−1.03	0.43	**0.02**	0.27	0.48	0.58
	Smoking—past	−0.15	0.33	0.64	0.52	0.50	0.30
	Smoking—current	−0.44	0.38	0.25	−0.68	0.38	0.08
**Zero**	IBD	1.84	0.37	**<0.01**	−1.40	0.44	**<0.01**
	Sex—male	0.32	0.24	0.18	−0.47	0.35	0.18
	Age	−0.02	0.01	**0.02**	0.01	0.01	0.37
	BMI—underweight	−0.03	0.68	0.97	−0.06	1.27	0.96
	BMI—overweight	0.46	0.34	0.17	0.01	0.43	0.98
	BMI—obese	0.88	0.78	0.26	−0.53	0.50	0.29
	Smoking—past	0.19	0.31	0.55	0.48	0.45	0.29
	Smoking—current	0.66	0.39	0.10	−0.42	0.39	0.27

^1^*Blastocystis* spp. includes *B. hominis* and *Blastocystis* sp. subtypes 1, 2, 3, and 4.

## Data Availability

Metagenomes and sample metadata for the 1000IBD study (https://1000ibd.org/, accessed on 14 August 2022) may be accessed subject to approval of the corresponding Data Access Committee at https://ega-archive.org/datasets/EGAD00001004194, accessed on 14 August 2022. The 500FG study (https://hfgp.bbmri.nl/menu/main/home, accessed on 14 August 2022) metagenomes and sample metadata are publicly available at https://www.ebi.ac.uk/ena/data/view/PRJNA319574, accessed on 14 August 2022. Metagenomes and sample metadata from the FMT study are publicly available at https://www.ncbi.nlm.nih.gov/bioproject/PRJNA625520/, accessed on 14 August 2022. The workflow for this study can be found at https://github.com/ginaguzzo/2021_gut_eukaryotes_in_IBD, accessed on 14 August 2022. As this study contains previously published datasets, we have adhered to the STORMS checklist where possible. The completed checklist for this research can be found at https://github.com/ginaguzzo/2021_gut_eukaryotes_in_IBD/blob/main/STORMS_Checklist.xlsx, accessed on 14 August 2022.

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
