# Peer review of "Individuals with Inflammatory Bowel Disease Have an Altered Gut Microbiome Composition of Fungi and Protozoa"

_microorganisms, 2022, doi:10.3390/microorganisms10101910_

Round 1
Reviewer 1 Report
In this paper, Guzzo et al., have characterized the eukaryoptic microbial communities using available shotgun stoll metagenomes from patients with IBD and healthy control. The study is well designed and balanced, and gives new insights into de importance of microbiome in IBD. However, there are some points that need to be clarified.
One of my mayor concerns is the difference in age between the two cohorts (IBD vs control). Although the microbiome remains relatively stable over adulthood, age could represent a bias in these types of studies. Changes over life should be considered (see: 10.1038/s41579-022-00768-z). This need to be, at least, addressed in the study.
Please, consider rewriting the section 3.3 Effect of faecal microbiota transplantation on eukaryotes. The description of the results is somewhat confusing. For example, in Fig 3 they represent the Sham group as success or failure although this group only received a saline solution. What do the authors mean with this?
Line 329. The authors claim that recipient 15 achieved successful engraftment but in the next sentence they say the opposite.
Line 333. This sentence insinuates that patient two and eight recieve FMT but they appear in the Sham group. Also, the authors mention figure 3A and B in the text but there is only on figure.
In section 3.4 Influence of cohort demographics on gut eukaryote composition the authors highlight the associations between Blastocystis with BMI and IBD. However, table 2 shows that such associations depend on the model used. Please, explain.
Line 382. "The IBD group in our study had a higher diversity and abundance of funggi compared to the healthy cohort". It is difficult to appreciate this statement in the figures 1 and 2.
Lines 467-469. The authors should be cautious with this type of statement, even more so it is based in one single sample from a study.
Author Response
Response for microorganisms-1894217, titled "Individuals with inflammatory bowel disease have an altered gut microbiome composition of fungi and protozoa"
We thank the reviewer for their recognition of our study design and its contributions to the study of the gut microbiome in IBD, and for providing their helpful and detailed feedback. We have addressed their specific comments as follows:
Specific comments:
- One of my mayor concerns is the difference in age between the two cohorts (IBD vs control). Although the microbiome remains relatively stable over adulthood, age could represent a bias in these types of studies. Changes over life should be considered (see: 10.1038/s41579-022-00768-z). This need to be, at least, addressed in the study.
Answer: The reviewer raises a valid concern regarding the age distribution between the two cohorts, and this was one of the reasons we included age as a covariate in our regression modelling. We have tried to address their concerns by adding the following section to the Discussion [lines 475-485]:
‘Unlike geography, we could not properly age-match the participants between our studies, and there was a notable age difference between the two cohorts (Table 1). It is known that age can affect the diversity and composition of the gut microbiome [82], including intestinal fungi which can differ in diversity and composition from infancy to adulthood [83]. However, little is known about how fungal populations change throughout adulthood as previous work has not compared older adults to younger ones [84,85]. In our study, we found that increased age in adulthood was associated with an increased abundance of S. cerevisiae (Table 2), and this trend may also occur in other fungi we observed. We also found that older adults were less likely to have the Blastocystis spp. present in their gut microbiome, indicating that protozoa should be a point of inclusion in future age-stratified research.’
We also thank the reviewer for bringing the citation to our attention and have added it in the above section (citation #82).
Additionally, we added age ranges to Table 1 to better illustrate the age distributions of the two cohorts.
- Please, consider rewriting the section 3.3 Effect of faecal microbiota transplantation on eukaryotes. The description of the results is somewhat confusing. For example, in Fig 3 they represent the Sham group as success or failure although this group only received a saline solution. What do the authors mean with this?
Answer: Our description and presentation of the results in section 3.3 and Figure 3 was based off the original study design by Kong et al. (2020) in which the FMT dataset was derived (doi: https://doi.org/10.1053/j.gastro.2020.08.045). The metric we used to define success or failure was also the primary metric used in the original publication, but we agree that it is confusing. We have redefined ‘success’ and ‘failure’ from another metadata category available from the original study: steroid-free clinical remission at week 10. We have updated Figure 3 with this metric, which has moved patients P2 and P9 into the FMT failure category.
In accordance with this change, we have also revised the sentence in section 2.1 (Study populations and design) to: ‘A successful outcome of FMT was defined by Sokol, et al. [36] as steroid-free clinical remission at 10 weeks post-FMT’ [lines 112-115]. And added this explanation to Figure 3’s legend: ‘Success refers to steroid-free clinical remission at ten weeks post-FMT’ [line 381].
Part of section 3.3 has been rewritten to the following [lines 349-364]:
‘Only one of the seven donor samples had a eukaryotic species present (Figure 3). There did not appear to be a relationship between the type of species, FMT group, and FMT outcome (measured as steroid-free clinical remission at 10 weeks post-FMT) of the recipients. Several FMT recipients—recipients one, four, nine, and 15—maintained the same eukaryotic species both before and after receiving donor FMT. These individuals all experienced remission for 10 weeks post-FMT, suggesting that some eukaryotes may not interfere with success following FMT. One eukaryotic species, Blastocystis sp. subtype 1, was also identified in a donor sample (donor 47) but this species was not present in their recipient's samples. Instead, their recipient (subject 15) maintained Penicillium roqueforti FM164 before and after FMT. Several fungal species including P. roqueforti FM164 were present in individuals with both success and failure outcomes—D. hansenii CBS767, P. roqueforti FM164, P. kudriavzevii, and S. cerevisiae S288C—which indicates a more complex relationship between intestinal fungi and FMT than can be resolved from this single, small study.’
- Line 329. The authors claim that recipient 15 achieved successful engraftment but in the next sentence they say the opposite.
Answer: This sentence has been removed.
- Line 333. This sentence insinuates that patient two and eight recieve FMT but they appear in the Sham group. Also, the authors mention figure 3A and B in the text but there is only on figure.
Answer: This sentence has now been removed since reformatting section 3.3 (see above). Additionally, ‘A’ and ‘B’ have been removed from the references to Figure 3.
- In section 3.4 Influence of cohort demographics on gut eukaryote composition the authors highlight the associations between Blastocystis with BMI and IBD. However, table 2 shows that such associations depend on the model used. Please, explain.
Answer: We have added lines to section 3.4 to address this:
‘Disease state was statistically associated with the prevalence of Blastocystis spp. but not the abundance of Blastocystis spp. (Table 2). This result indicates that only the presence of Blastocystis spp. is linked to a healthy state, and within those who have Blastocystis spp., the abundance of this genus does not statistically differ between individuals with or without IBD’ [lines 397-401].
- Line 382. "The IBD group in our study had a higher diversity and abundance of funggi compared to the healthy cohort". It is difficult to appreciate this statement in the figures 1 and 2.
Answer: This statement has been revised to specifically illustrate the main differences in fungi between the two cohorts: ‘The IBD group in our study had a higher prevalence and abundance of fungi—notably Saccharomyces, Nakaseomyces, Debaryomyces, and Candida species—compared to the healthy cohort (Figure 2)’ [lines 424-426].
- Lines 467-469. The authors should be cautious with this type of statement, even more so it is based in one single sample from a study.
Answer: We agree that we should be careful with statements like this and have changed this line to read, ‘Interestingly, the recipient achieved a successful FMT outcome, which further supports investigating whether Blastocystis spp. are indeed safe to donate in FMT [72]’ [lines 538-541].
References
Kong, L, Lloyd-Price, J, Vatanen, T, Seksik, P, Beaugerie, L, Simon, T, Vlamakis, H, Sokol, H & Xavier, RJ 2020, 'Linking Strain Engraftment in Fecal Microbiota Transplantation With Maintenance of Remission in Crohn's Disease', Gastroenterology, vol. 159, no. 6, Dec, pp. 2193-2202.e2195.
Reviewer 2 Report
This is the first comprehensive investigation of intestinal eukaryotes using whole metagenome shotgun sequencing data in a significant number of inflammatory bowel disease (IBD) patients. The authors reported a higher diversity and abundance of fungi in IBD patients compared to the healthy cohort.
Comment
The authors are to be commended for their excellent and novel work. I had also read their review published in the Journal Inflammatory Bowel Diseases earlier this year (Inflamm Bowel Dis 2022 Jul 1;28(7):1112-1122), and this work is the result of extensive research and effort using the most cutting edge techniques. Although there is still work to do concerning finding the way to intervene in the gut microbiome in order to tackle inflammation processes in IBD, this work raises concerns over the use of fecal microbiota transplantation and certain probiotics such as Saccharomyces Cerevisiae Boulardii in IBD patients. The authors could add a short comment in their Discussion over the use of such probiotic supplements in IBD patients, because we, as clinicians, need to be aware of potential adverse events affecting the disease course and avoid using them. Please put probiotic supplements under scrutiny in your future research, as they are generally considered as "safe" or sometimes even necessary.
Author Response
Response for microorganisms-1894217
We thank the reviewer for commending our work, and for acknowledging our previous work that inspired this study on an often-neglected population of intestinal microbes.
The reviewer raises an interesting point regarding S. cerevisiae and the common probiotic strain S. cerevisiae var. boulardii. The reviewer’s comment has brought to our attention that some strain-level identifications from our results were mistakenly not included in our study. We have added these details to Figures 2 & 3, as well as in the Results section and Figure legends.
In the case of S. cerevisiae, only S. cerevisiae S288C was identified in our study. However, this is the only S. cerevisiae strain available in the EukDetect database, and Khatri et al. (2017) (doi: https://doi.org/10.1038/s41598-017-00414-2) showed that both S. cerevisiae S288C and boulardii share over 99% genome sequence similarity. Thus, even with this strain-level identification, it is still worth raising a point on the safety of probiotic strains.
We have added the following lines to our Discussion to address this point:
‘The fact that S. cerevisiae was more prevalent in individuals with IBD may have implications for microbial-based therapies, such as probiotics. A popular probiotic S. cerevisiae strain, S. cerevisiae var. boulardii, has been trialled for various gastrointestinal disorders [65] and is generally regarded as safe, though there have been several reports of adverse events [66-68]. This strain has been trialled for IBD as well, but its safety and efficacy for the disease remain inconclusive [69]. While we only detected the strain S. cerevisiae S288C in our study, that may be because it was the only strain of S. cerevisiae included in the EukDetect database. The two strains share over 99% genome sequence similarity [70] and, therefore, individuals with IBD may be more susceptible to colonisation or adverse effects of S. cerevisiae strains. More research is needed to understand the eukaryotic microbiome in IBD to ensure the safety of eukaryotic probiotic strains such as S. boulardii.’ [lines 456-467].
References
Khatri, I, Tomar, R, Ganesan, K, Prasad, GS & Subramanian, S 2017, 'Complete genome sequence and comparative genomics of the probiotic yeast Saccharomyces boulardii', Sci Rep, vol. 7, no. 1, 2017/03/23, p. 371.
Reviewer 3 Report
- Use italics for fungi in the abstract, too
- Report p values comparing case and controls in Table 1
- % of smoker in Table 1 seem wrong
- In Table 2 report at least 1 digit different from “0” in p values
- “All participants were recruited in France.” “We controlled for this confounding effect by using samples from the same country (Netherlands) in our study”
France or Netherlands?
Author Response
We thank the reviewer for their corrections and feedback. We have addressed their specific comments below.
Specific comments:
1. Use italics for fungi in the abstract, too
Answer: The fungal species have now been italicised in the abstract [lines 25-26].
2. Report p values comparing case and controls in Table 1
Answer: We respectfully disagree with the reviewer on this point because Table 1 in our study is only for descriptive purposes and not intended to make any inferences (see: https://medium.com/peter-flom-the-blog/do-not-put-p-values-in-your-table-1-8aad0d6c92d). In the case where inferences were to be made, then p-values would be appropriate. Our Table 1 was guided by Hayes-Larson et al. (2019) (doi: https://doi.org/10.1016%2Fj.jclinepi.2019.06.011) who also do not recommend adding p-values to Table 1.
3. % of smoker in Table 1 seem wrong
Answer: We believe the reviewer may be referring to the percentage of current and past smokers in the healthy control cohort, which have the same percentage. These values were based off the first version of the 500FG metadata which was indeed incorrect, and we have updated these to the current version available on the database (https://hfgp.bbmri.nl/menu/500fgdata/dataexplorer?entity=FG500_public_data).
Due to the updated metadata formatting, we have changed both the IBD and control groups to have three categories for smoking status: ‘current’, ‘past’, and ‘never’ (Table 1) and have re-run the regression analysis with these updated categories (Table 2). The workflow for the updated metadata can be found here: https://github.com/ginaguzzo/2021_gut_eukaryotes_in_IBD/blob/main/R_analysis/00_formatting_cohort_metadata.R
4. In Table 2 report at least 1 digit different from “0” in p values
Answer: We have changed “0.00” in Table 2 to “< 0.01”.
5. “All participants were recruited in France.” “We controlled for this confounding effect by using samples from the same country (Netherlands) in our study”
France or Netherlands?
Answer: These two statements were referring to different parts of the analysis; the individuals from France were in the FMT study, whereas the individuals from Netherlands were in the two cohort studies that were compared. We have edited both statements to make this clearer.
The first section has been changed to, ‘The dataset included 115 longitudinal shotgun stool metagenome samples from 17 adults with CD pre- and post-FMT, and single timepoint samples from their healthy FMT donors (n = 5), originally published in Kong, et al. [12]. All participants in this study were recruited in France’ [lines 101-105].
The second statement now reads, ‘We controlled for this confounding effect by comparing two cohorts from the same country (Netherlands) in part of our study, and this comparison was kept separate from our analysis of the individuals in the FMT study who were recruited in France’ [lines 494-497].
References
Hayes-Larson, E, Kezios, KL, Mooney, SJ & Lovasi, G 2019, 'Who is in this study, anyway? Guidelines for a useful Table 1', J Clin Epidemiol, vol. 114, Oct, pp. 125-132.
Round 2
Reviewer 1 Report
I would like to thank the authors for addressing my concerns. The manuscript has increased its strength and the research gives a new approach regarding the microbiome in IBD that is usually disregarded. Congratulations on this nice work.